# Rational design of complex phenotype via network models

**Marcio Gameiro** [1,2☯] *, **Tomáš Gedeon** [3☯], **Shane Kepley** [1☯], **Konstantin Mischaikow** [1☯]

**1** Department of Mathematics, Rutgers, The State University of New Jersey, Piscataway, New Jersey, United States of America, **2** Instituto de Ciências Matemáticas e de Computação, Universidade de São Paulo, São Carlos, São Paulo, Brazil, **3** Department of Mathematical Sciences, Montana State University, Bozeman, Montana, United States of America

☯ These authors contributed equally to this work.
* gameiro@math.rutgers.edu

**Data Availability Statement:** The data and code used in this work are available via Github at https://github.com/marciogameiro/three-node-hysteresis.

**Funding:** The work of M.G., S.K., and K.M. was partially supported by the National Science

## Abstract

We demonstrate a modeling and computational framework that allows for rapid screening of thousands of potential network designs for particular dynamic behavior. To illustrate this capability we consider the problem of hysteresis, a prerequisite for construction of robust bistable switches and hence a cornerstone for construction of more complex synthetic circuits. We evaluate and rank most three node networks according to their ability to robustly exhibit hysteresis where robustness is measured with respect to parameters over multiple dynamic phenotypes. Focusing on the highest ranked networks, we demonstrate how additional robustness and design constraints can be applied. We compare our results to more traditional methods based on specific parameterization of ordinary differential equation models and demonstrate a strong qualitative match at a small fraction of the computational cost.

## Author summary

A major challenge in the domains of systems and synthetic biology is an inability to efficiently predict function(s) of complex networks. This work demonstrates a modeling and computational framework that allows for a mathematically justifiable rigorous screening of thousands of potential network designs for a wide variety of dynamical behavior. We screen all 3-node genetic networks and rank them based on their ability to act as an inducible bistable switch. Our results are summarized in a searchable database that can be used to construct robust switches. The ability to quickly screen thousands of designs significantly reduces the set of viable designs and allows synthetic biologists to focus their experimental and more traditional modeling tools to this much smaller set.

## 1 Introduction

Ever since the dawn of cellular biology, the central analogy that we employ to describe cells is that of miniature machines that transform the information about its environment to

Foundation under awards DMS-1839294 and HDR TRIPODS award CCF-1934924, DARPA contract HR0011-16-2-0033, and National Institutes of Health award R01 GM126555. K.M. is also supported by a grant from the Simons Foundation. The work of M.G. was also partially supported by FAPESP grant 2019/06249-7 and by CNPq grant 309073/2019-7. The work of T. G. was partially supported by NSF grant DMS-1839299, DARPA FA8750-17-C-0054 and NIH 5R01GM126555-01. The funders had no role in study design, data collection and analysis, decision to publish, or preparation of the manuscript.

**Competing interests:** The authors have declared that no competing interests exist.

appropriate responses. The responses take the form of increased or decreased gene expression, protein activation or deactivation, or regulation of transport between cellular compartments and exterior of the cell. There is only a short step from viewing cells as little machines to the desire of controlling them, repairing them, and then building new cellular functions. This is the starting point of synthetic biology [1–4]; for recent review of progress and challenges see [5]. The success of engineered mechanical or electronic systems, crucially depends on (i) modularity of their designs and (ii) ability to model complicated assemblies of parts before they are built. In synthetic biology both of these steps present significant challenges. The focus of this contribution is on a novel mathematical approach to addressing the second challenge.

The strength of our approach that we call Dynamic Signatures Generated by Regulatory Networks (DSGRN) is that we are agnostic to the specific biochemical or biophysical design of the elements of the circuits that we analyze. The input consists of a mathematical abstraction of a gene regulatory network, e.g. Fig 1a, that consists of nodes and annotated directed edges indicating activation or repression. The user is required to provide a means of scoring the behavior of the network from the information about dynamics that is computed by DSGRN. The DSGRN software [6] then allows a ranking of the networks in question based on this score.

To demonstrate the applicability of DSGRN we focus on the question of design of a hysteretic switch. There are three reasons that make this a natural choice. First, it is conceptually simple. The same ideas can be applied to the design or analysis of more complicated logic circuits, but this naturally entails a corresponding increase in complexity of computation and analysis. Second, it is one of the early successes of synthetic biology [7]. Third, its resolution requires a global understanding of the dynamics of the design over multiple phenotypes, e.g. monostability versus multistability, and therefore is a nontrivial mathematical problem. While experimental implementation of a design consisting of two mutually repressing transcription factors was a great triumph of predictive modeling, the design of [7] seems to be fragile and follow-up attempts [8] have been made to make it more robust. A natural question arises if more complex networks are able to exhibit a more robust switching behavior. This paper provides an efficient algorithmic approach towards addressing this question.

We begin with the well established observation that in vivo gene regulatory networks operate under noisy conditions [9, 10]. Rather than attempting to provide a specific model for the noise, we adopt the perspective that noise is significant enough to impact the initial conditions of the dynamics and the parameter values at which the network operates, but not so significant that it overwhelms the underlying nonlinear dynamics. Thus, for this paper we adopt the following *design principle*: a synthetic network should attempt to maximize the range of the phase space and parameter space where it exhibits the desired function.

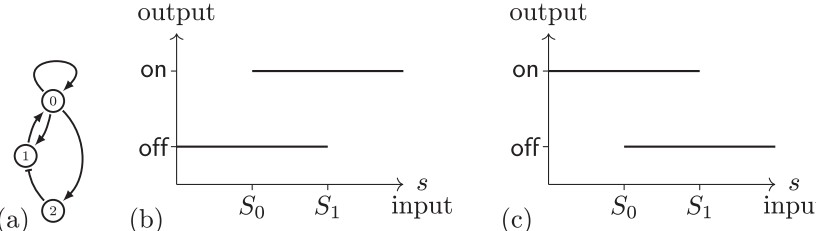

**Fig 1.** (a) Regulatory network 107. There are three nodes labelled 0, 1, and 2. Edges → indicates up regulation and ⊣ indicates down regulation. (b) Conceptual image of ascending hysteresis. Off/low output for low values of input signal $s$. On/high output for high values of $s$. Bistability for intermediate values of $s$ allowing for hysteresis. (c) Descending hysteresis. Results, analogous to those for ascending hysteresis, are discussed in Section 5.

Fig 1b summarizes the minimal structure and functionality of an (ascending) hysteretic switch. As values of an input signal are increased from a low level, the output signal is off. Once the input signal achieves a given threshold, $S_1$, the output signal changes to on and remains at on for high values of the signal. As the input signal is lowered the output value remains on until the input signal reaches the threshold $S_0$, that is lower than $S_1$, at which point the output signal switches to off.

This property of "remembering" past states is called *hysteresis*. Since on and off are determined by dynamics, they must be represented by stable states. To obtain hysteresis requires that both stable states be present for the range of signal between $S_0$ and $S_1$, e.g. that the system exhibits bistability. For this reason a system of this type is often referred to as a *bistable switch*.

In order to have a well defined problem we ask and provide answers to the following question: Which three-node networks exhibit the functionality of a bistable switch over the largest range of parameter values? The reader is no doubt aware that as of yet we have not described our model for the network dynamics nor indicated what signals indicate on and off.

This is discussed in varying detail in later sections. We have adopted this approach in an attempt to emphasize that DSGRN can be used with minimal knowledge of the rather substantial mathematical theory and machinery that justifies the software [11–17]. While dynamics is expressed as an action on a phase space, the specific action very much depends on parameters. With this in mind the DSGRN model provides a combinatorial representation of a decomposition of parameter space and combinatorial representations of dynamics. We attempt in Section 2 **I1**-**I3** to provide a minimal description of these combinatorial representations that allows us to describe our results. We do not expect that this description is sufficient for the typical reader to understand how DSGRN works. Thus, we provide more detailed descriptions of various aspects of the DSGRN machinery in Section 4.

We remark that it is the fact that DSGRN is a combinatorial model that allows us to perform extremely efficient computations, and as indicated above, allows DSGRN to be agnostic to the biochemical or biophysical details. Of course, it is precisely these details that play essential roles in the actual construction of components of a synthetic network. With this in mind, the true novelty of DSGRN is that it employs ideas from computational algebraic geometry to provide an explicit decomposition of parameter space [15, 17] on which dynamics is understood. It is unreasonable to expect that these precise bounds on parameters should be valid for more traditional models involving explicit nonlinearities. Nevertheless, as we demonstrate in the context of ordinary differential equation (ODE) models using Hill function nonlinearities with more than 20 dimensional parameter spaces, DSGRN provides considerable insight into parameter values at which bistable switching occurs.

## 2 Results

Our goal is to identify three-node networks that act as bistable switches over large regions of parameter space. As indicated in Fig 1a we label the nodes in our network by 0, 1, and 2, and assume that node 0 is directly affected by the input and the output of the network is expressed via node 2. Since each node can influence any other node (itself included) in three ways—activation, repression, or no impact—there are $3^9 = 19,683$ distinct three node networks. The stipulation that 0 is an input node and 2 is the output node precludes any reduction in the number of networks due to symmetries. We exclude 5, 103 trivial networks as defined in Section 4.6.

The number of regions into which DSGRN decomposes parameter space grows rapidly with the number of edges in the network. For example, three-node networks with 8 edges can have up to 823, 011, 840 distinct parameter regions, while for 9 edges this number increases to 93, 329, 542, 656. Because of this size, we only consider one network with 9 edges, that where

each edge is an activator. Therefore, we consider 14, 068 networks. These networks are analyzed using the DSGRN software described in Section 4. However, for the purpose of reporting the results we include the following information about the combinatorial structure of DSGRN. **I1** provides information about the combinatorial dynamics. **I2** and **I3** discuss the decomposition of parameter space.

I1.   For three node networks the phase space is $\{(x_0, x_1, x_2) \mid x_n > 0\}$ where the variable $x_n$ is associated with node $n$. For a given parameter value DSGRN decomposes phase space into cubes defined by the hyperplanes $x_n = \theta_{m,n}$ where $\theta_{m,n}$ is the threshold parameter associated with an edge from node $n$ to node $m$. The global dynamics at the given parameter value is determined by a *state transition graph (STG)* defined on these cubes. A cube $C$ that has a self edge under the STG is labeled as an $\mathtt{FP}(i_0, i_1, i_2)$. This should be interpreted as a stable state under the associated dynamics. The $i_k \in \{0, 1, 2, 3\}$ indicates that if $x \in C$, then $x_k$ is greater than $i_k$ of the thresholds $\theta_{*,k}$, and thus provides information about the location in phase space of the stable state.

I2.   The parameter space for the DSGRN model consists of multiple positive real numbers associated with each node (1 for the node, 2 for each incoming edge, and 1 for each outgoing edge). For node $n$ the DSGRN software produces a finite decomposition of parameter space and encodes this decomposition via a *factor graph*, denoted by $PG(n)$. Two vertices in the factor graph are connected by an edge if they represent regions of the continuous parameter space whose closures intersect on a codimension-one face. Details about the parameters are presented in Section 4.1. For the moment we remark that if at a vertex in the factor graph parameters associated with the in-edges do not align properly (i.e. the parameters corresponding to the in-edges are consistently too high, or consistently too low) with the parameters associated with the out-edges, then one can remove edges associated with the node. This in turn implies that the dynamics is captured by a simpler regulatory network. A node in the factor graph is defined to be *essential* if every in-edge and every out-edge are relevant for the dynamics [16]. The essential factor graph $PG_e(n)$ is the subgraph of the factor graph $PG(n)$ consisting of the essential nodes.

I3.   The full parameter space of a regulatory network is a product of the parameter spaces associated with each node. The decomposition of the full parameter space is indexed by the *parameter graph PG*. Since each region of this decomposition is made up of the product of the region from the decomposition of the parameter space of each node, the parameter graph is the product of the factor graphs, i.e. $PG = \prod_{n=0}^{2} PG(n)$. Of fundamental importance is the fact that for each node in the parameter graph the state transition graph is constant over all parameters in the associate region. For each node in the parameter graph the DSGRN output includes the $\mathtt{FP}(i_0, i_1, i_2)$ that arise from the associated state transition graph.

Fix a regulatory network with a fixed set of parameter values. In particular, this identifies a unique vertex $(v_1, v_2)$ in the graph $PG(1) \times PG(2)$. Since we are interested in a direct correspondence between network topology and bistable switching we restrict our attention to the essential nodes $PG_e(1) \times PG_e(2)$.

We view the continuous change of the input signal $s$ to node 0 as a curve through the parameter space associated with node 0, e.g. monotone change in inducer concentration induces a monotone change in abundance of protein produced by gene 0 (cf. IPTG in [7]).

The DSGRN analogue to a continuous change in the inducer is a discrete path $v_0^0, \ldots, v_0^t, \ldots, v_0^T$ within the factor graph $PG(0)$, which is realized in the entire parameter

graph as a path $(v_0^0, v_1, v_2), \ldots, (v_0^t, v_1, v_2), \ldots, (v_0^T, v_1, v_2)$ within the graph $PG(0) \times (v_1, v_2)$. Each vertex $(v_0^t, v_1, v_2)$ on this path is an element of the parameter graph $PG$ and hence for each vertex DSGRN can determine the global dynamics.

We say the path exhibits *ascending hysteresis* if at the initial vertex of the path $(v_0^0, v_1, v_2)$ there is a $\mathtt{FP}(i_0, i_1, j_1)$, at the final vertex of the path $(v_0^T, v_1, v_2)$ there is a $\mathtt{FP}(i_0, i_1, j_3)$ where $j_3 > j_1$, and at some intermediate vertex of the path there are two stable states $\mathtt{FP}(i_0, i_1, j_1)$ and $\mathtt{FP}(i_0, i_1, j_2)$ with $j_2 > j_1$. For the purposes of this paper we set $j_1 = 0$ and require that $j_2 > 0$ and $j_3 > 0$. Note that since we need to observe at least three distinct forms of global dynamics we insist that our paths be of length at least three.

Since such a path need not traverse all of $PG(0)$ we refer to it as a *partial path*. We focus on partial paths because we do not presume to know the parameter values associated to node 0 at which the regulatory network is acting in the absence of the input signal (cf. in the context of construction of the toggle switch [7] we do not presume to know the level of protein production in the absence of the added IPTG).

We define the *hysteresis score* of a regulatory network to be the percentage of paths that exhibit ascending hysteresis among all paths. The total number of the paths is given by the number of paths of length at least 3 in $PG(0)$ times the total number of vertices in $PG_e(1) \times PG_e(2)$.

The ranking according to the hysteresis score is presented in Fig 2 (left). Observe that the typical three-node network is incapable of exhibiting hysteresis and less than 1% of networks are capable of producing hysteresis for the majority of parameter values. However, fourteen networks are capable of producing hysteresis for more than 60% of the paths. Based on our design principle we now restrict (for the most part) our attention to these fourteen three-node regulatory networks shown in Fig 3.

Returning to the motivation of our design principle that in vivo gene regulatory networks operate under noisy conditions, we remark that by restricting our analysis to essential nodes we are assuming that even under noisy conditions each edge of the regulatory network operates effectively. For example, networks 1-4 in Fig 3 have a partial path hysteresis score of 100% since they exhibit partial path hysteresis in all of their essential parameter nodes. However, if

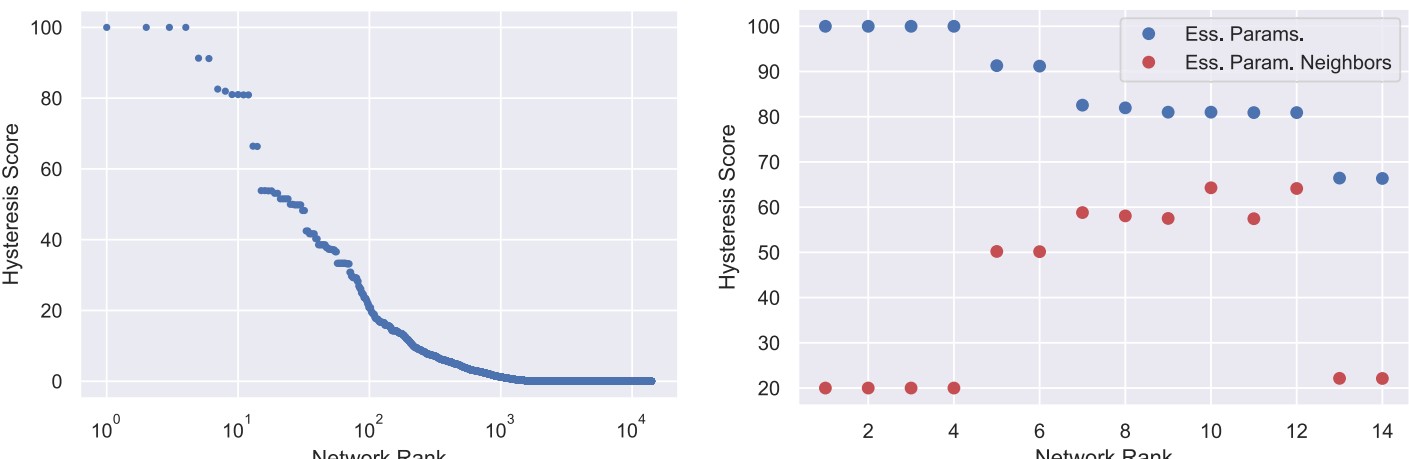

**Fig 2.** Left: 14,098 three node networks ranked by (ascending) hysteresis score analyzed at the essential parameters. This is the percentage of partial paths in the essential parameter sub-graph which exhibit switch-like behavior. The rapid decrease in score indicates that most networks are unlikely to exhibit hysteresis at most parameters. Right: The top scoring networks are further discriminated by scoring hysteresis in a neighborhood of the essential parameters. The scores after perturbation (red) are significantly lower than the scores at essential parameters (blue) for the top 4 networks. This is an indication that these networks are fragile i.e. will fail to perform well if any component of the network is removed.

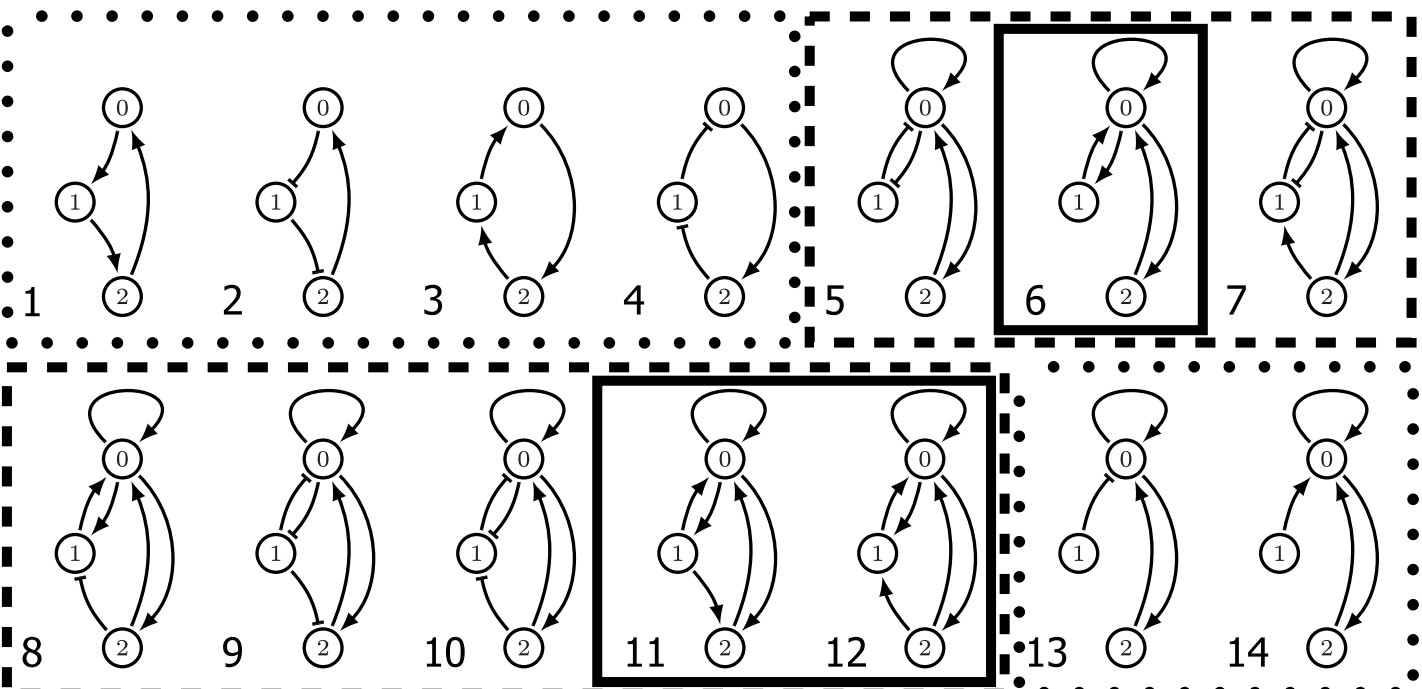

**Fig 3. The top fourteen regulatory networks by hysteresis score.** The six regulatory networks in dotted boxes are fragile, while the eight outlined via the dashed lines are robust. Observe that the three networks outlined by the solid lines are also consistent and each node acts only as an activator.

any one of the edges is removed, then the remaining network will not be bistable and not capable of hysteresis (we return to this point in greater detail in Section 4.4). This observation motivates a search for a measure of the robustness of hysteresis with respect to network perturbations.

A more reasonable assumption might be that not all edges in the regulatory network function effectively at all times. To capture this, for the top fourteen regulatory networks we consider the set of parameter nodes in $PG(1) \times PG(2)$ that are within one edge of $PG_e(1) \times PG_e(2)$ and repeat the computation of the partial path hysteresis score. The results—we call this the *perturbed hysteresis score*—are shown in red in Fig 2 (right).

As expected, the perturbed hysteresis score is less than the hysteresis score. However, this loss of functionality varies widely across networks and is difficult to predict from the topology alone. We define the *robustness score* of a regulatory network to be its perturbed hysteresis score divided by its hysteresis score. We refer to each network by its position in the list ordered by decreasing hysteresis score (see [18]). Networks 1-4, 13, and 14 (boxed with dotted lines in Fig 3) have robustness scores under 0.5. This suggests that under ideal conditions these networks will perform well as a switch. However, they are easy to break in the sense that small perturbations from the essential parameters largely destroy their ability to act as a switch. With this in mind we call regulatory networks with robustness score less than or equal to 0.5 *fragile*, while those scoring above 0.5 are referred to as *robust* (and are boxed by dashed lines in Fig 3).

Invoking the hysteresis rank and robustness allows us to reduce our attention to eight regulatory networks at which point we can focus on the actual topology of the design. The implementation of a gene regulatory network is constrained by available control mechanisms which must be considered when comparing networks. For instance, Network 5 in Fig 3 requires that node 0 act both as an activator and repressor. While this is biologically possible, e.g. the dimer CI acting in phage λ lysogenic/lytic switch, simpler design features may be desired. We call a

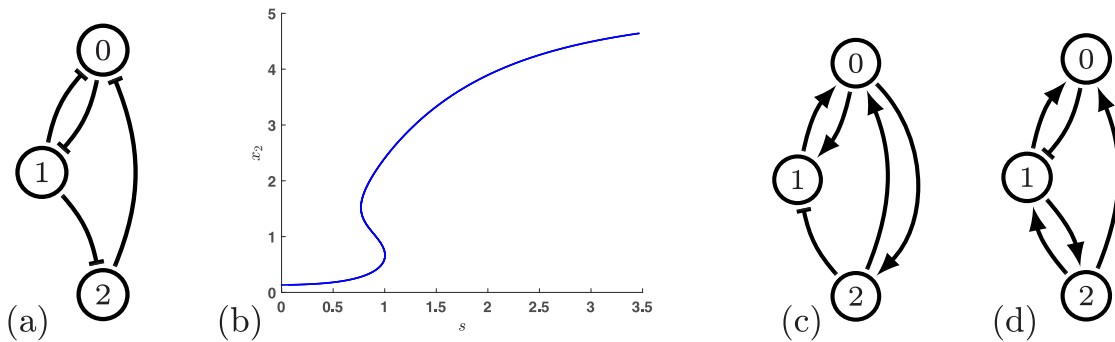

**Fig 4.** (a) The best network with only repressing edges, network 66, has a hysteresis score of 33.33%. (b) Continuation of equilibria in the Hill model ([1]) for regulatory network 12 using $n = 4$. (c) Regulatory network 33. (d) Regulatory network 3839.

node *consistent* if it acts as an activator or a repressor, but not both. As is indicated by the solid boxes in Fig 3 there are three high ranked and robust regulatory networks in which all nodes are consistent: 6, 11, and 12.

Note that in networks 13 and 14, node 1 provides a constant input to node 0. Therefore node 1 does not affect the existence of ascending hysteresis. Removing this ineffectual node transforms networks 13 and 14 into a two node toggle switch with mutually activating edges, and with positive self-regulation on node 0. The fact that these networks are fragile in our analysis recapitulates the observation from [7, 8] that the two-node design of the toggle switch is fragile.

It is interesting to observe that the top three consistent regulatory networks that provide ascending hysteresis are based on nodes that are activators. The consistent regulatory network based on repressing nodes that has the highest hysteresis score (33.33%) is shown in Fig 4. This is a fragile network with the perturbed hysteresis score of 6.66%.

Thus, simple robust design of ascending hysteresis seems to require the use of activators. Interestingly, the role of activators in ascending hysteresis is not mirrored by the role of repressors in the descending hysteresis. First, no 3-node network that only consists of activators is capable of producing descending hysteresis. Second, in contrast with Fig 3, there is no network among the top 14 networks for descending hysteresis, with only repressing edges (see Fig 9).

Based on three criteria—hysteresis score, robustness score, and consistency of nodes—Network 12 is the most desirable design. Returning to the question posed in the introduction—are more complex networks capable of exhibiting more robust switching behavior—the answer is a qualified yes. However, complexity alone is not sufficient. This is evidenced by the fact that if Network 12 is modified by adding an additional activating edge from node 1 to node 2, a self activation for either 1 or 2, or any combination of these edges the resulting network has a smaller hysteresis score, often dramatically so. For instance if we attempt to maximize complexity by adding every single edge as an activator, the resulting network has a hysteresis score of 0%.

The results discussed up to this point have all been obtained from the combinatorial computations of DSGRN. More traditional modeling of regulatory networks is based on ODEs. As is discussed in Section 4.1 and in Section 6, there is a direct translation from DSGRN parameters to nonlinearities based on Hill functions in the limit when the exponents in the Hill function are very large. We now demonstrate that information from DSGRN has implications for the ODE models. Two important observations are that (i) trustworthy ODE computations are many orders of magnitude more expensive than DSGRN computations, and (ii) it is

unreasonable to expect the explicit DSGRN decomposition of parameter space to apply precisely to any specific ODE.

To expand on this we consider Network 12 and a corresponding ODE. Assumptions need to be made on how multiple in-edges to a node impact the rate of change of the associated variable. These assumptions are discussed in detail in Section 4.1, but for the moment it suffices to state that since all the arrows in Network 12 have the form → leads to a summation of the nonlinear terms affecting the growth rate. Hence we consider

$$
\begin{aligned}
\dot{x}_0 &= -\gamma_0 x_0 + L_0 + \frac{\delta_{0,0} x_0^n}{\theta_{0,0}^n + x_0^n} + \frac{\delta_{0,1} x_1^n}{\theta_{0,1}^n + x_1^n} + \frac{\delta_{0,2} x_2^n}{\theta_{0,2}^n + x_2^n} + s \\
\dot{x}_1 &= -\gamma_1 x_1 + L_1 + \frac{\delta_{1,0} x_0^n}{\theta_{1,0}^n + x_0^n} + \frac{\delta_{1,2} x_2^n}{\theta_{1,2}^n + x_2^n} \\
\dot{x}_2 &= -\gamma_2 x_2 + L_2 + \frac{\delta_{2,0} x_0^n}{\theta_{2,0}^n + x_0^n}
\end{aligned}
\tag{1}
$$

where for simplicity have made two modeling assumptions. First, the effect of the external signal on the growth rate of $x_0$ is given by a simple linear additive term $s$. Second, the exponents $n$ of the Hill functions are the same. In addition, there are 21 other parameters that lie in $(0, \infty)^{21}$ (see Section 4.1). There are 707 vertices in $PG(0)$ (see [15]), and 24 vertices in $PG_e(1) \times PG_e(2)$ (see Section 4.2).

The most computationally efficient means of identifying the desired hysteresis curve in Eq (1) is to fix a parameter value in $(0, \infty)^{22}$, choose an initial value $s_0$ for $s$, find a stable fixed point with low $x_2$ value, perform continuation with respect to arc-length of a fixed length, and check that two saddle-node bifurcations have occurred. An example of this computation is shown in Fig 4b (see Section 6 for details). With the goal of quantifying how robustly this system exhibits hysteresis the obvious question is how many parameter values should be chosen and what is the appropriate choice of arc-length. Based on the number of vertices in $PG(0)$ and $PG_e(1) \times PG_e(2)$, the number of partial paths computed by DSGRN is on the order of $10^5$. Furthermore, since each region of parameter space is an unbounded open set in $(0, \infty)^{22}$ even sampling each region is non-trivial. This suggests that performing sufficiently many continuation computations to compare with the DSGRN hysteresis score is prohibitively expensive.

With this in mind we greatly simplify the DSGRN computations being performed. We remark that a partial order can be placed on the vertices of $PG(0)$ (see Section 4.1) such that there is a unique minimal vertex $\underline{v}_0$ and unique maximal vertex $\bar{v}_0$. A path $v_0^0, \ldots, v_0^t, \ldots, v_0^T$ within the factor graph $PG(0)$ is *full* if $v_0^0 = \underline{v}_0$ and $v_0^T = \bar{v}_0$. This leads to two new scores obtained as follows.

For each vertex in $(v_1, v_2) \in PG_e(1) \times PG_e(2)$ we consider all full paths $(\underline{v}_0, v_1, v_2), \ldots, (v_0^t, v_1, v_2), \ldots, (\bar{v}_0, v_1, v_2)$ and mark those paths that exhibits hysteresis as hysteretic. The *full path hysteresis score* is the percentage of hysteretic paths among all full paths. The *perturbed full path hysteresis score* is the same but based on the one edge neighborhood of $PG_e(1) \times PG_e(2)$.

To compare the predictions of DSGRN against ODE models we chose four regulatory networks, Network 12 from Fig 3, Network 107 from Fig 1a, and Network 33 and 3839 shown in Fig 4c and 4d. These latter three networks were chosen because they exhibit different hysteresis scores: 42.46%, 18.95%, and 0%, respectively. For each network we performed two sets of experiments. For the first we randomly chose 1000 parameter values that lay in $\underline{v}_0 \times PG_e(1) \times PG_e(2)$, and for the second we chose 1000 parameter values in the one edge neighborhood with respect to $PG_e(1) \times PG_e(2)$. In each case for each parameter choice we

**Table 1. The regulatory network number comes from the ranking of the hysteresis score [18].** The bottom two rows indicate the full path and partial path hysteresis scores obtained from DSGRN. The first column indicates the exponent of the Hill function used in the ODE model for the regulatory network, e.g. (1). The two columns under each regulatory network number indicate the percentage of continuation computations that result in a hysteresis curve. The first column assumes the parameter value is in a region defined by $\underline{v}_0 \times FP(1) \times FP(2)$ and the second column assume the parameter value is in region defined by a one edge neighborhood. 1000 curves were computed for each entry.

| Regulatory Network | 12 | | 33 | | 107 | | 3839 | |
|---|---|---|---|---|---|---|---|---|
| Hill function exponent $n$ | Hysteresis Score | Perturbed Score | Hysteresis Score | Perturbed Score | Hysteresis Score | Perturbed Score | Hysteresis Score | Perturbed Score |
| 30 | 96.4% | 72.2% | 84.8% | 34.5% | 29.7% | 57.1% | 6.8% | 3.8% |
| 20 | 92.2% | 58.1% | 78.5% | 30.2% | 16.7% | 42.9% | 7.3% | 4.5% |
| 10 | 68% | 26.3% | 50% | 16.9% | 2.8% | 16.1% | 7.8% | 3.6% |
| 5 | 17.7% | 3.6% | 12.4% | 3.4% | 0% | 2.3% | 7.5% | 2.1% |
| 4 | 8.9% | 1.6% | 6.1% | 1.4% | 0% | 0.5% | 4.4% | 1.2% |
| DSGRN (full path) | 100% | 79.09% | 83.33% | 61.67% | 33.96% | 25.05% | 0% | 0% |
| DSGRN (partial path) | 80.91% | 64.13% | 42.46% | 27.73% | 18.95% | 13.34% | 0% | 0% |

performed the above mentioned procedure to identify whether or not one obtains a hysteresis curve. The results are indicated in Table 1.

We highlight three observations from Table 1.

- *For large Hill exponent n the DSGRN full path scores and the ODE scores are quite similar.* This is not surprising. DSGRN is based on a mathematical approach to nonlinear dynamics that captures features that persist under perturbation [11–14, 19].

- *For more biologically realistic levels of n the quantitative agreement between the scores disappears.* Again, this is not surprising. It has long been known that in order for nonlinearities with gentle sigmoidal shape to intersect at multiple points, their parameters must be carefully adjusted. As a result, for low *n*, bistability is rare.

- *The relative ranking by DSGRN of the capability of regulatory networks to achieve robust switching is predictive of the observations from the ODE models.* Moving from left to right along the rows, DSGRN predicts that the corresponding networks are progressively less capable of acting as a robust switch. For the most part the ODE simulations agree with this prediction. Most importantly, Network 12 is the best at all values of *n*. We include Networks 107 and 3839 to emphasize that the predictive power of DSGRN is not perfect. However, for these networks the realization of ascending hysteresis is consistently low, again suggesting that DSGRN is capable of identifying regulatory networks of interest.

## 3 Discussion

DSGRN provides a modeling framework and associated computational tool that is capable of analyzing all 3-node regulatory networks for prevalence over a large range of parameter values of a particular phenotype. Our investigation into the identification of the robust expression of the phenotype of hysteresis demonstrates DSGRN's practical value—in synthetic biology hysteresis forms a basis for a design of a bistable switch. It also demonstrates the power of DSGRN to capture complex dynamics—hysteresis arises from global organization of multiple phenotypes (monostability, bistability, monostability) as a function of increasing external input. Furthermore, the publicly available searchable database of all 3-node networks allows synthetic biologist to select robust designs that meets additional implementation criteria [18].

It is important to note that the complexity of hysteresis phenotype makes it challenging to succinctly describe the network features i.e. number, sign and position of edges, that characterize high scoring networks. While it is known that presence of positive edges generally leads to bistability, our computations show that there is no simple relationship between the hysteresis score and the number of positive edges. Furthermore, we believe that as size and complexity of networks increase, simple network features are even less likely to predict presence or absence of specific dynamics. Thus, a direct evaluation of the prevalence of such dynamics across parameters by DSGRN becomes a crucial tool in understanding of behavior of complex networks.

Obviously, DSGRN can be used to search for simpler phenotypes. In particular, it has been used to catalog types and number of intermediate steady states in epithelial-mesenchymal transition network [20], as well as to characterize START network controlling G1/S transition in human cell cycle [16]. In principle it can be applied to the analysis of more complicated control circuits.

DSGRN occupies a novel niche in the collection of modeling tools for regulatory networks. On one hand it is similar to Boolean models, where in the simplest setting gene expression is either on or off, i.e. 0 or 1, and the update rule that encodes the dynamics is a Boolean function. The dynamics of Boolean models is thus efficiently computable. Conceptually, the closest analogue to DSGRN is the work pioneered by L. Glass and S. Kauffman [21, 22] involving *switching systems* where the logic of the Boolean system is embedded into continuous differential equations with the goal of predicting qualitative features of differential equation dynamics by the dynamics of the asynchronously updated embedded Boolean system. The state transition graphs used by DGSRN extend the embedded Boolean systems and allow for modeling a broader class of dynamics, while preserving the efficiency of computations. DSGRN also combinatorializes parameter space to understand how dynamics changes under the change in parameters. Again, similar to the Boolean models the goal of DSGRN is not to precisely match and reproduce carefully measured expression data of genes over a wide variety of growth conditions.

However, in the setting of systems biology more often than not such measurements are not available, especially for networks involving more than a few genes. In such situations, DSGRN can be a first step in understanding of network dynamics. DSGRN can search through many proposed networks over a wide range of parameter values, and eliminate those that do not support the desired dynamical behavior, coarsely defined e.g. equilibria or oscillations with particular patterns of high and low expression values. Elimination of networks or reduction of potential functional parameter values for a given network provides significant reduction of hypotheses space.

In contexts where one has carefully measured expression data of genes, modeling tools of choice often involve ODEs with experimentally determined parameters. In contrast to the Boolean approach, the mathematical foundations of DSGRN—a continuous phase space and parameter space—allows for direct comparison with an extremely broad class of ODE models [19]. As is demonstrated in this paper, DSGRN provides a means to compare systems of ODEs. In particular, it can rank the relative ability of ODE models to produce particular dynamics over large ranges of parameter values. At the same time, DSGRN provides a priori bounds on parameter regions where sampling of parameters and fitting the expression data is feasible. Finally, DSGRN provides, at low computational cost, the ability to describe relationships between any simultaneous change in many parameters and the changes in network dynamics. This facilitates generation of hypothesis of behavior of a system under different conditions and leads to prioritization of experiments.

To add additional emphasis on the importance of the computational efficacy of DSGRN we note that the comparisons in Table 1 are based on full path and perturbed full path hysteresis scores. We expect that in many applications it is more likely that external control will not lead to a path that extends across the entire parameter domain of the input node. In this case the partial path statistics are more relevant. However, carrying out such computations in the setting of tradition ODE models appears to be computationally prohibitive.

Due to its ability to describe complex relationship between network parameters and network dynamics, albeit on a coarse level, and the associated systematic reduction of the hypothesis space for experimental examination of this dynamics, DSGRN should become a part of an essential toolbox in systems and synthetic biology.

## 4 Methods

We provide a brief description of how DSGRN combinatorializes both phase space and parameter space of regulatory networks. For more details the reader is referred to [15, 16, 23].

### 4.1 Input and output

DSGRN takes as input an annotated directed graph (see Fig 3), called a *regulatory network*, where the annotations on the edges indicate activation → or repression ⊣ along with an algebraic expression that indicates how incoming edges to a node interact. To understand the role of algebraic expression we note that implicit in the DSGRN calculations is a positive variable $x_n$, e.g. level of protein, associated with node $n$ of the regulatory network. Each $x_n$ decays at a rate $\gamma_n$. If there is an edge from node $m$ to node $n$, then the model includes three positive parameters: $\ell_{n,m}$, a low growth rate of $x_n$ induced by $x_m$; $\delta_{n,m}$, such that $\ell_{n,m} + \delta_{n,m}$ represents a high growth rate of $x_n$ induced by $x_m$; and $\theta_{n,m}$, a *threshold* that separates the values of $x_m$ that induce low or high growth rate of $x_n$. In particular, if node $n$ has a single in-edge → from $m$ then the increase or decrease of $x_n$ is determined by the sign of

$$-\gamma_n x_n + \begin{cases} \ell_{n,m} & \text{if } x_m < \theta_{n,m} \\ \ell_{n,m} + \delta_{n,m} & \text{if } x_m > \theta_{n,m}. \end{cases} \tag{2}$$

If there are multiple in-edges to node $n$, then the user has considerable flexibility in deciding whether to add or multiply the rates associated with the in-edges. For this paper we adopted the convention to first add rates associated with → edges, and then multiply by values associated with ⊣ edges. In particular, because Network 12 consists exclusively of → edges, all the nonlinearities are summed (this in turn leads to the form of (1)). In the case of Network 33 (see Fig 4c) the nonlinearities that drive the production of $x_1$ would be multiplied, i.e.

$$-\gamma_1 x_1 + \left( \begin{cases} \ell_{1,0} & \text{if } x_0 < \theta_{1,0} \\ \ell_{1,0} + \delta_{1,0} & \text{if } x_0 > \theta_{1,0} \end{cases} \right) \left( \begin{cases} \ell_{1,2} + \delta_{1,2} & \text{if } x_2 < \theta_{1,2} \\ \ell_{1,2} & \text{if } x_2 > \theta_{1,2}. \end{cases} \right)$$

Given a regulatory network as input DSGRN is capable of producing as output a queryable database, called the *DSGRN database*, indicating the possible global dynamics at associated parameters. Conceptually it is useful to view the DSGRN database via the *parameter graph* (described below), where associated to each node in the parameter graph is an explicit region in parameter space and a description of the global dynamics in the form of a *Morse graph* (described below). For a fixed ordering of the thresholds (see below), an edge between two nodes in the parameter graph indicates that the associated regions share a co-dimension 1 boundary.

Finally, we remark that there is an apparent symmetry relating the topology of a network and the algebraic expressions which govern the interactions between its nodes. The simplest example can be seen in Networks 1-4 which have very similar topology. Each has exactly 3 edges which connect the nodes cyclically. One also notices that they have identical hysteresis and robustness scores so that, in some sense, these networks are "dynamically equivalent". A related symmetry for STGs has been identified and studied in [24]. In [24], all observable patterns of fixed points and cycles were enumerated and classified for 3 node networks (assuming no self edges). Each distinct pattern was identified with a corresponding Boolean 3-cube with directed edges in a specific configuration and the dynamically equivalent configurations were related by permutations of the 3-cube.

The similar topologies and scores for Networks 1-4 in this work can be attributed to the fact that for these networks, this STG symmetry is also preserved along paths through the DSGRN parameter graph. In fact, for Networks 1-4, one can essentially prove this equivalence "by hand". However, we do not exploit this symmetry in this work because, outside of the simplest cases such as Networks 1-4, this symmetry is not well understood despite being easily observed [17]. Obtaining a deeper understanding of the relationship between a generic network's topology and the algebraic expressions governing the interactions between its nodes is an open problem.

## 4.2 Parameter graph

As indicated above, given a regulatory network with $N$ nodes and $E$ edges, the DSGRN parameter space is $(0, \infty)^{N+3E}$. The parameter graph provides combinatorial representation of a finite decomposition of this parameter space. Each node of the parameter graph corresponds to an explicit open semi-algebraic set [15, 17] with the property that the STG (see Section 4.3) is constant for all parameters in that set.

As is discussed in **I2** and **I3** the parameter graph is the product of the factor graphs and each factor graph is determined by the in-edges and out-edges of its corresponding node in the regulatory network. Fig 5a shows the factor graphs for nodes whose number of in and out-edges are (from left to right) (2, 1), (1, 2), and (1, 1).

General descriptions of how parameters are identified with nodes of the factor graph can be found in [15, 17]. To provide intuition we focus on the simplest factor graph corresponding to (1, 1), i.e. one in-edge and one out-edge.

Because there is a single in-edge and a single out-edge, as indicated in (2) there is a unique $\gamma$, $\theta$, $\ell$ and $\delta$. Furthermore, the sign of the expression (2) is constant over the subsets of parameter space defined by the inequalities

$$\gamma_n\theta_{k,n} < \ell_{n,m} < \ell_{n,m} + \delta_{n,m}, \quad \ell_{n,m} < \gamma_n\theta_{k,n} < \ell_{n,m} + \delta_{n,m}, \quad \ell_{n,m} < \ell_{n,m} + \delta_{n,m} < \gamma_n\theta_{k,n} \quad (3)$$

where $\theta_{k,n}$ is the threshold associated with the edge from node $n$ to node $k$. These three regions are represented by the nodes in the rightmost factor graph $PG(2)$ in Fig 5a.

Observe that if the parameters satisfy the leftmost or rightmost sets of inequalities in (3), then the sign of (2) is independent of the value of $x_m$. Since we are assuming that $n$ has a unique in- and out-edge, this implies that we can remove node $n$ from the network without losing information about the potential dynamics (the constant growth rate on $x_k$ due to $x_n$ will be compensated for by the parameter values). Thus only the node associated with the middle set of inequalities in (3) is essential, as is indicated by the blue node in Fig 5a.

We remark that in the context of switching systems the analogue of an essential parameter node is the notion of *effective regulator* [25]. However, the restriction in the DSGRN setting is

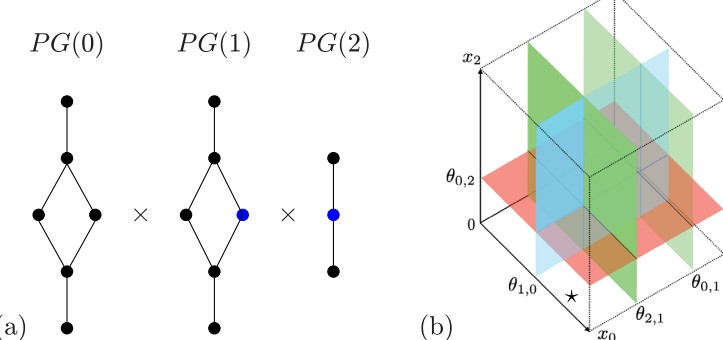

**Fig 5.** (a) For fixed ordering of the thresholds the parameter graph for the network in Fig 4a where the nodes 0, 1, and 2 have 2 in-edges and 1 out-edge,1 in-edge and 2 out-edges, and 1 in-edge and 1 out-edge, respectively, written as a Cartesian product of the three factor graphs. For $i = 1, 2$, $PG_e(i)$ is the subgraph consisting of only the blue vertices. (b) Phase space decomposition for Regulatory Network shown in Fig 4a under the assumption that $\theta_{2,1} < \theta_{0,1}$. The cell identified by $\star$ is labeled (1, 0, 0).

in the choice of parameter, i.e. a node in the parameter graph, as opposed to the choice of a Boolean function defined on the regulatory network.

Observe that the order of the set of inequalities of (3) can be obtained by associating it with increasing values of $\gamma$. This same approach applies in general and we use it to induce a partial order on any form of factor graph that $PG(0)$ may assume. All full and partial paths discussed in Section 2 are chosen to be strictly monotone with respect this partial order.

### 4.3 Combinatorial dynamics

Note that if there are $s_n$ out-edges from node $n$, there must be $s_n$ thresholds associated to node $n$ with indices of the form, $\theta_{*,n}$, and these thresholds divide the domain of the variable, $x_n$, into $s_n + 1$ intervals. This in turn implies that for a given regulatory network with $N$ nodes there is a decomposition of the phase space $(0, \infty)^N$ into rectangular cells bounded by thresholds, zero, or extending to infinity.

Each cell is labeled by a vector $(\alpha_0, \alpha_1, \ldots, \alpha_{N-1})$, $\alpha_n \in \{0, \ldots, s_n\}$, where $s_n \in \{0, \ldots, N\}$ is the number of out-edges of node $n$ in the network under consideration. See Fig 5b for an illustration for a three node network i.e. $N = 3$.

The combinatorial dynamics is represented by a STG as defined in [15] and Section 7. Each node of the STG represents one rectangular cell and edges indicate how cells are mapped forward in time.

We conclude this section by emphasizing that as presented in [15], DSGRN was not capable of analyzing networks with self repressing interactions or nodes without an out-edge. As part of this work we remove these restrictions (see Section 7 for details). The DSGRN code is available at [6].

### 4.4 Intuition into robustness and fragility

A focus of this paper is on identifying regulatory networks that, if they can be built, will perform as desired under a variety of settings. This led to a measure of robustness and fragility. We do not claim to have a sharp characterization of the quantities, but we can provide a posteriori intuition.

To understand fragility consider Network 1 in Fig 3. Because there is one out-edge for each node, there is one hyperplane $x_n = \theta_{m,n}$ associated to each coordinate of phase space $(0, \infty)^3$.

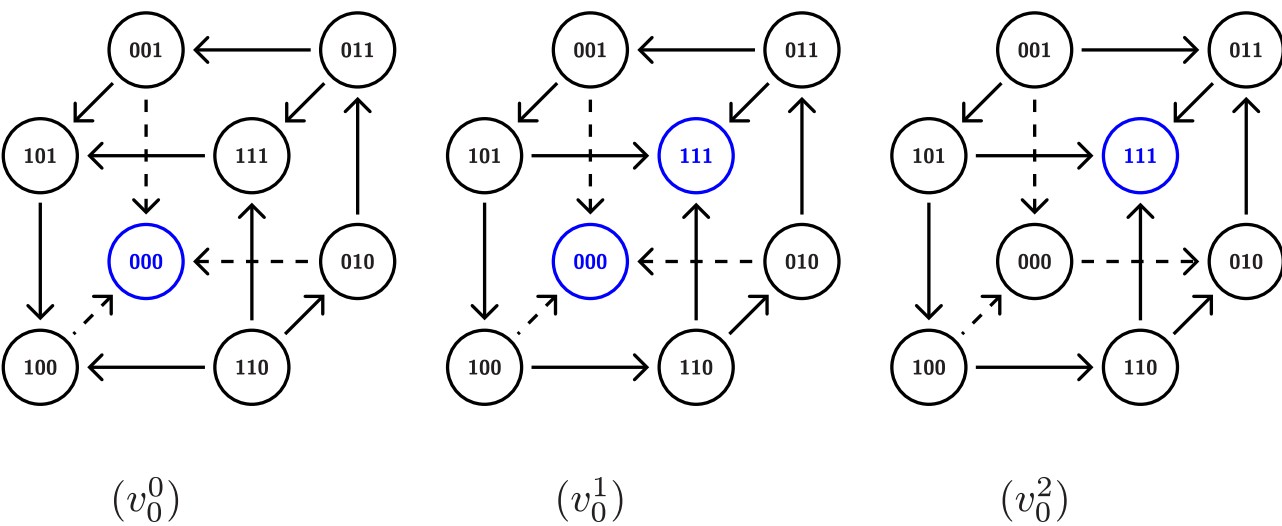

**Fig 6. Hysteresis representation in DSGRN.** State transition graph for Network 1 in Fig 3 over a factor graph $PG(0)$ at the unique essential node of $PG$ $(1) \times PG(2)$. $(v_0^0)$ represents a low input to node 0 and exhibits $\mathtt{FP}(0, 0, 0)$; $(v_0^1)$ represents a medium input to node 0 and exhibits bistability between $\mathtt{FP}$ $(0, 0, 0)$ and $\mathtt{FP}(1, 1, 1)$; $(v_0^2)$ represents a high input to node 0 and exhibits $\mathtt{FP}(1, 1, 1)$.

Thus phase space is divided into eight three-dimensional cubes indexed by $\{0, 1\}^3$ that is represented by the graph shown in Fig 6 where each node represents a cube and edges indicate that the two associated cubes intersect along a hyperplane. For this simple example, the direction of the arrow is determined by the sign of (2) evaluated at the hyperplane $x_n = \theta_{m,n}$ (see Section 7 and [15] for the general procedure).

There are 27 nodes in the parameter graph for Network 1 in Fig 3. The same argument as presented in Section 4.2 shows that there is a single essential node in $PG_e(1) \times PG_e(2)$ given by the inequalities

$$\ell_{1,0} < \gamma_1 \theta_{2,1} < \ell_{1,0} + \delta_{1,0} \quad \text{and} \quad \ell_{2,1} < \gamma_2 \theta_{0,2} < \ell_{2,1} + \delta_{2,1}. \tag{4}$$

Again, as discussed in Section 4.2, the factor graph for $PG(0)$ consists of three nodes and thus for Network 1 there is a unique full and unique partial path $v_0^0, v_0^1, v_0^2$. The STG associated to each node in the path through parameter the parameter graph are indicated in Fig 6.

The three STGs shown in Fig 6 indicate the existence of ascending hysteresis. The blue nodes in the STGs indicate the attracting states that, as indicated in **I1**, DSGRN labels as an $\mathtt{FP}$. Thus moving from left to right we have monostability ($\mathtt{FP}(0, 0, 0)$), bistability ($\mathtt{FP}(0, 0, 0)$ and $\mathtt{FP}(1, 1, 1)$), and monostability ($\mathtt{FP}(1, 1, 1)$). Observe that the $x_2$ values at these attracting states (again moving from left to right) are 0, 0 and 1, and 1. Since this is the unique partial path for Network 1, the hysteresis score is 100%, in agreement with Fig 2. However, we leave it to the reader to check that if one chooses a node in $PG(1) \times PG(2)$ that differs from the essential node by a single inequality (there are four such nodes), then along the associated path one will not achieve the desired bistability state. Thus, the perturbed hysteresis score is 20% and Network 1 is labeled as fragile.

To provide intuition into robustness consider Network 6 in Fig 3. Both node 1 and node 2 have a single in and out-edge, and thus there is a single essential node in $PG(1) \times PG(2)$. Having fixed this parameter value, we need to consider the STGs associated with paths over the factor graph $PG(0)$. Observe that phase space is partitioned into regions bounded by the hyperplanes defined by $x_0 = \theta_{1,0}$, $x_0 = \theta_{2,0}$, $x_0 = \theta_{3,0}$, $x_1 = \theta_{0,1}$, and $x_2 = \theta_{0,2}$. Thus the nodes of the desired STGs are as shown in Fig 7.

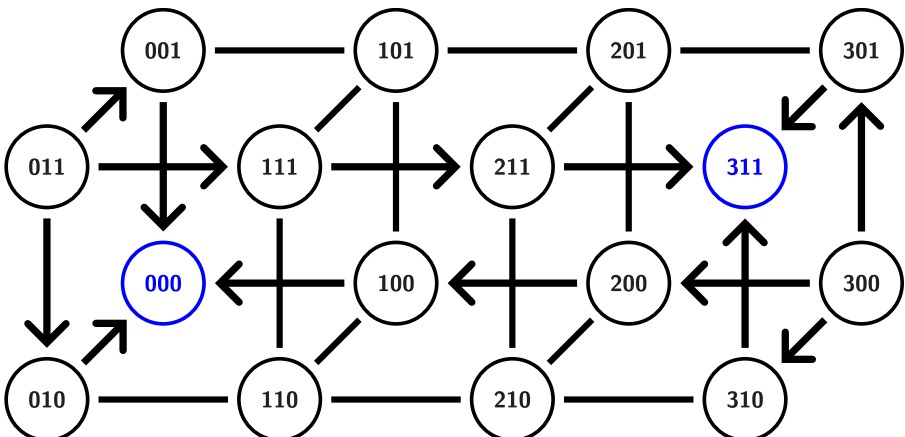

**Fig 7. STG for network 6 from Fig 3.** Since node 0 has three output edges there are three thresholds of the variable corresponding to 0 and four states 0,1,2,3. The arrows are valid for any essential parameter nodes in *PG*. The direction of other, un-oriented edges, depends on a choice of a particular essential node in *PG*(0). Since the bistabilty between (000) and (311) is assured for any such choice resulting in lack of fragility of the bistablty and hysteresis.

We begin by focusing on identifying bistability. High values of variables $x_1$ and $x_2$ are represented by the four cubes labeled (0, 1, 1), (1, 1, 1), (2, 1, 1), and (3, 1, 1) where $x_1 > \theta_{0,1}$ and $x_2 > \theta_{0,2}$. Observe that for these cubes the variable $x_0$ will increase (arrows pointing to the right). Similarly, low values of $x_1$ and $x_2$ are represented by (0, 0, 0), (1, 0, 0), (2, 0, 0), and (3, 0, 0) and there the variable $x_0$ will decrease (arrows pointing to the left). Let us now restrict our attention to essential nodes in *PG*(0). We leave it to the reader to check that for any essential node in *PG*(0) the direction of arrows on the left and right squares are as depicted in Fig 7. The directions of the other, unoriented edges, are dependent upon the specific essential node. However, observe that the bistabilty between (000) and (311) is assured for any such choice. As a consequence of this multitude of means of maintaining bistability, perturbing away from the essential node of $PG_e(1) \times PG_e(2)$ does not necessarily destroy bistability. Finally, consider any full path over the factor graph *PG*(0). We claim that at the endpoints of this graph the STG gives rise to monostability. However, any such path goes through an essential node of *PG*(0) and thus experiences bistability. Similarly, there are full paths over the factor graph *PG*(0) based at nodes obtained from perturbing away from the essential node of $PG(1) \times PG(2)$. Therefore, it is not surprising that this network exhibits robust hysteresis. Again, we emphasize that this is an a posteriori computation; we can explain the results found from the DSGRN computations, but we cannot predict them.

We remark that there are similarities and differences in the concept of robustness used in this paper from those that are explored in the context of switching systems [25, 26] or Boolean models [27]. The overwhelming similarity is that we are concerned with whether the dynamics observed at one parameter value is equivalent to that at nearby parameter values. In our case parameter space is continuous and partitioned into a finite set of regions. Thus, nearby parameters either lie in the same region or a region that differs by a co-dimension one hypersurface. In the case of the Boolean models [27], a nearby parameter value is a Boolean function that differs by a single entry. A subtle difference is that our primary focus is not on matching the existence of individual trajectories arising as solutions to a differential equation that to the temporal sequence of the Boolean updates, but rather with the existence of a global dynamical structures, e.g. monostability and bistability. A more important difference is that we are interested in tracking and organizing these global structures over large ranges of parameter space,

e.g. hysteresis consists of a prescribed combinatorial sequence of monostability, bistability, monostability.

## 4.5 Morse graphs

The information in the STG is summarized by a *Morse graph*. This is an acyclic directed graph, or, equivalently, a partially ordered set, where nodes indicate potential recurrent dynamics and the directed edges indicate the direction of the dynamics between recurrent sets [15, 28]. We summarize the importance of Morse graph representation of dynamics by noting that any *minimal* node of the Morse graph labeled $FP(\alpha_0, \alpha_1, \alpha_2)$ indicates that the corresponding cell is an attracting region for the dynamics [15]. Thus a Morse graph with a unique minimal node suggests monostability, while two minimal nodes indicates bistability.

## 4.6 Constraints on searched networks

There are $3^9 = 19,683$ three node networks. We only consider a subset of these defined as follows. Let $a_{ij} \in \{-1, 0, 1\}$ denote the *edge coefficients* which describe the type (or lack) of interaction from node $j$ to node $i$. Specifically, $a_{ij} = 0$ if there is no interaction, $a_{ij} = -1$ if node $j$ represses node $i$, and $a_{ij} = 1$ if node $j$ activates node $i$. We say that a three node network is *trivial* if either there exists no path from node 0 to node 2, or no path from node 1 to node 2. In terms of the edge coefficients, a network is trivial if and only if

$$|a_{20}a_{21}| + |a_{20}a_{01}| + |a_{21}a_{10}| = 0.$$

Our restriction to nontrivial networks follows from the observation that if a network has no path from node 0 to node 2, then it is incapable of acting as a switch. Furthermore, if there is no path from node 1 to node 2, then node 1 has no influence on the dynamics, and therefore can not be responsible for any hysteresis or lack thereof. We omit these 5, 103 trivial networks from our analysis.

As indicated in the introduction we omit all but one of the three node networks in which every gene interacts directly with every other gene. The remaining 14, 068 are the networks analyzed in this paper.

## 4.7 Computations

The computations of the DSGRN ascending hysteresis score presented in Fig 2 took a total wall time of 1, 478.61 hours. The computations were performed on a cluster with 100 nodes and finished after just over 15 hours. The numerical continuation for Hill models presented in Table 1 were computed on a single laptop and took a total time of 7.9 hours.

## 5 Results for descending hysteresis

We consider *descending hysteresis* in which the switch-like behavior transitions from a high steady state to a low steady state with bistability in between. A schematic for this case is shown in Fig 1c.

We carried out the same analysis as for the ascending hysteresis. The combinatorial definition of descending hysteresis is analogous to that of ascending hysteresis. Specifically, a path exhibits descending hysteresis if at the initial vertex of the path $(v_0^0, v_1, v_2)$ there is a $\mathrm{FP}(i_0, i_1, j_1)$, at the final vertex of the path $(v_0^T, v_1, v_2)$ there is a $\mathrm{FP}(i_0, i_1, j_3)$ with $j_3 < j_1$, and at some intermediate vertex of the path there are two stable states $\mathrm{FP}(i_0, i_1, j_1)$ and $\mathrm{FP}(i_0, i_1, j_2)$ with $j_2 < j_1$. For the purposes of this paper we set $j_3 = 0$ and require that $j_1 > 0$ and $j_2 > 0$. As in the study of the ascending hysteresis, we only consider paths of length at least three.

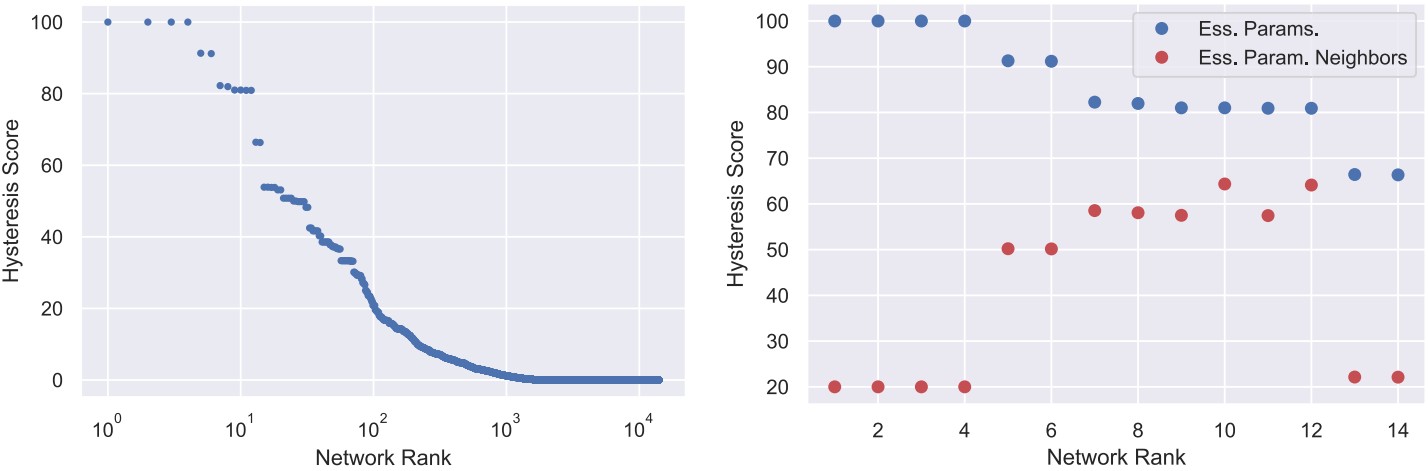

**Fig 8.** Left: 14,098 three node networks ranked by (descending) hysteresis score analyzed at the essential parameters. Right: The top scoring networks are further discriminated by scoring hysteresis in a neighborhood of the essential parameters. The scores after perturbation are shown in (red) and the scores at essential parameters in (blue).

The ranking according to the hysteresis score is presented in Fig 8 (left) along with the scores after perturbation for the top 14 networks in Fig 8 (right). These 14 networks are shown in Fig 9. Observe the striking similarity with Fig 3. In both cases there are exactly 4 networks that have a 100% hysteresis score and contain only 3 edges that cyclically connect the nodes. All of these networks are fragile and their corresponding robustness scores are also similar. Comparing Fig 3 with Fig 8 we observe that the distribution of ascending and descending hysteresis scores are very similar and indicate that robust switching is relatively rare in either case. The close similarity between analysis of the entire collection of 3 node networks with respect

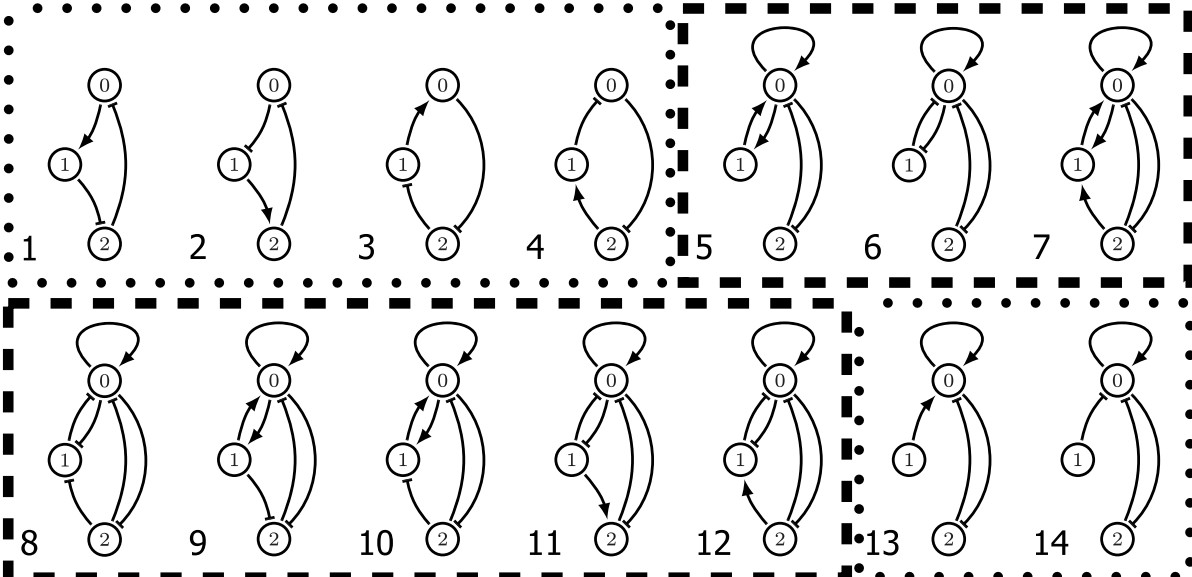

**Fig 9. The top fourteen regulatory networks by descending hysteresis score.** The six regulatory networks in dotted boxes are fragile, while the eight outlined via the dashed lines are robust. Observe that none of the networks are consistent. Networks 13 and 14 are analogous to networks 13 and 14 of the main paper. The fact that they are fragile recovers observation from [7] that 2-node toggle switch is fragile (see comment in the main text).

to ascending and descending hysteresis suggests that there exists some relation, e.g. symmetry, where a network's ascending hysteresis and robustness scores are comparable to its partner's descending scores. This relation does not appear to be obvious as it must be compatible with observations about asymmetry made in the main text. For example, even though there is a network consisting of only repressors that exhibits ascending hysteresis ranked networks, there is no network with only activators that exhibits descending hysteresis. Exploring this relationship and its implications is the subject of current research and remains an open problem.

## 6 Hill model continuation computations

The dynamics of a regulatory network as modelled by DSGRN is obtained by assuming that the rate of change of $x$ can be approximated by

$$-\Gamma x + \Lambda(x), \tag{5}$$

where $x = (x_0, \ldots, x_{N-1})$ represent the state variables of the $N$ nodes of the network, $\Gamma$ is a diagonal matrix

$$\Gamma = \begin{pmatrix} \gamma_0 & & & \\ & \gamma_1 & & \\ & & \ddots & \\ & & & \gamma_{N-1} \end{pmatrix}$$

where $\gamma_i > 0$ is the decay rate of $x_i$, and $\Lambda(x) = (\Lambda_0(x), \ldots, \Lambda_{N-1}(x))$ takes the form described below. Let

$$\sigma^+(y, \ell, \delta, \theta) = \begin{cases} \ell, & \text{if} \quad y < \theta \\ \ell + \delta, & \text{if} \quad y > \theta \end{cases}$$

and

$$\sigma^-(y, \ell, \delta, \theta) = \begin{cases} \ell + \delta, & \text{if} \quad y < \theta \\ \ell, & \text{if} \quad y > \theta. \end{cases}$$

Assume that node $i$ has $k$ in-edges from the nodes $j_1, \ldots, j_k$ in the regulatory network. Furthermore, assume that of these edges $j_1, \ldots, j_{k_1}$ are activating and $j_{k_1+1}, \ldots, j_k$ are repressing. For the computations of this paper we set

$$\Lambda_i(x) = (\sigma^+(x_{j_1}) + \cdots + \sigma^+(x_{j_{k_1}}))\sigma^-(x_{j_{k_1+1}}) \cdots \sigma^-(x_{j_k}). \tag{6}$$

As an example, for network 12 in Fig 3, Eq (5) takes the form

$$-\gamma_0 x_0 + \sigma^+(x_0, \ell_{0,0}, \delta_{0,0}, \theta_{0,0}) + \sigma^+(x_1, \ell_{0,1}, \delta_{0,1}, \theta_{0,1}) + \sigma^+(x_2, \ell_{0,2}, \delta_{0,2}, \theta_{0,2})$$
$$-\gamma_1 x_1 + \sigma^+(x_0, \ell_{1,0}, \delta_{1,0}, \theta_{1,0}) + \sigma^+(x_2, \ell_{1,2}, \delta_{1,2}, \theta_{1,2})$$
$$-\gamma_2 x_2 + \sigma^+(x_0, \ell_{2,0}, \delta_{2,0}, \theta_{2,0})$$

since all the in-edges are activating.

**Table 2. (Hysteresis and perturbed hysteresis scores) Results for comparisons with four networks.** Numbers for Regulatory network come from ranking in Fig 8. Bottom rows are the full and partial path hysteresis scores obtained from DSGRN for Regulatory networks 12, 33, 108, and 4346. The first column indicates the exponent of the Hill function used in the ODE model for the regulatory network. The two columns under each regulatory network number indicate the percentage of continuation computations that result in a hysteresis curve. The first column assumes the parameter value is in a region defined by $\underline{v}_0 \times FP(1) \times FP(2)$ and the second column assume the parameter value is in region defined by a one edge neighborhood. 1000 curves were computed for each entry.

| Regulatory Network | 12 | | 33 | | 108 | | 4346 | |
|---|---|---|---|---|---|---|---|---|
| Hill function exponent $n$ | Hysteresis Score | Perturbed Score | Hysteresis Score | Perturbed Score | Hysteresis Score | Perturbed Score | Hysteresis Score | Perturbed Score |
| 30 | 81.2% | 51.7% | 84.4% | 41.2% | 57.9% | 56.1% | 0% | 0% |
| 20 | 70.8% | 41.3% | 74.9% | 34.0% | 45.4% | 46.8% | 0% | 0% |
| 10 | 39.7% | 18.8% | 45.3% | 16.6% | 18.2% | 21.8% | 0% | 0% |
| 5 | 7.3% | 2.1% | 7.6% | 2.2% | 1.3% | 2.4% | 0% | 0% |
| 4 | 3.1% | 0.6% | 2.2% | 0.5% | 0.2% | 0.3% | 0% | 0% |
| DSGRN (full path) | 100% | 79.1% | 83.3% | 61.7% | 33.9% | 25.1% | 0% | 0% |
| DSGRN (partial path) | 80.9% | 64.1% | 42.5% | 27.7% | 18.9% | 13.3% | 0% | 0% |

For the numerical computations to obtain the results in Tables 1 and 2 we consider Hill function models

$$\dot{x}_0 = -\gamma_0 x_0 + H_0(x) + s$$
$$\dot{x}_1 = -\gamma_1 x_1 + H_1(x)$$
$$\dot{x}_2 = -\gamma_2 x_2 + H_2(x)$$

where $H_i$ is obtained from the regulatory network by replacing the step functions $\sigma^-$ and $\sigma^+$ in $\Lambda_i$ by the decreasing and increasing Hill functions

$$H^-(x, \ell, \delta, \theta, n) = \ell + \delta \frac{\theta^n}{\theta^n + x^n}$$

and

$$H^+(x, \ell, \delta, \theta, n) = \ell + \delta \frac{x^n}{\theta^n + x^n},$$

respectively. We represent the input signal to node 0 by the additive parameter $s$ in the first equation.

Returning to network 12 in Fig 3, the Hill model is given by

$$\dot{x}_0 = -\gamma_0 x_0 + \ell_{0,0} + \frac{\delta_{0,0} x_0^n}{\theta_{0,0}^n + x_0^n} + \ell_{0,1} + \frac{\delta_{0,1} x_0^n}{\theta_{0,1}^n + x_1^n} + \ell_{0,2} + \frac{\delta_{0,2} x_2^n}{\theta_{0,2}^n + x_2^n} + s$$

$$\dot{x}_1 = -\gamma_1 x_1 + \ell_{1,0} + \frac{\delta_{1,0} x_0^n}{\theta_{1,0}^n + x_0^n} + \ell_{1,2} + \frac{\delta_{1,2} x_2^n}{\theta_{1,2}^n + x_2^n}$$

$$\dot{x}_2 = -\gamma_2 x_2 + \ell_{2,0} + \frac{\delta_{2,0} x_2^n}{\theta_{2,0}^n + x_0^n}.$$

Notice that in (1) we combined the $\ell$ parameters as $L_0 = \ell_{0,0} + \ell_{0,1} + \ell_{0,2}$, $L_1 = \ell_{1,0} + \ell_{1,2}$, and $L_2 = \ell_{2,0}$.

Using the Hill models we compute curves of equilibria using a pseudo arclength continuation method [29] to detect fold bifurcation points. A sample continuation curve for (1) is presented in Fig 4b, where the following values of parameters were used:: $\gamma_0 = 1$, $\gamma_1 = 1$, $\gamma_2 = 1$,

$L_0 = 0.508736659464953$, $L_1 = 0.823149364604282$, $L_2 = 0.129562882298977$, $\theta_{0,0} = 2.742699202456864$, $\theta_{1,0} = 3.176067131107269$, $\theta_{2,0} = 3.406985767928092$, $\theta_{0,1} = 1.753260803421655$, $\theta_{0,2} = 0.724695975751957$, $\theta_{1,2} = 1.566020932246446$, $\delta_{0,0} = 1.172412555847297$, $\delta_{1,0} = 2.862607698545040$, $\delta_{2,0} = 4.947150771599252$, $\delta_{0,1} = 0.946904335902318$, $\delta_{0,2} = 0.077108238106769$, and $\delta_{1,2} = 1.624416688203425$.

For the numerical continuation computations DSGRN provides sample parameter values from parameter regions and for each sampled parameter point we search for hysteresis using the following procedure.

For ascending (descending) hysteresis we randomly sample 10 initial guesses $(x_0^0, x_1^0, x_2^0)$ satisfying the conditions $0 < x_0^0 \leq \min \theta_{*,0}$, $0 < x_1^0 \leq \min \theta_{*,1}$, and $0 < x_2^0 \leq \min \theta_{*,2}$ ($\max \theta_{*,2} < x_2^0 \leq 2 \max \theta_{*,2}$ for descending hysteresis). For each initial guess $(x_0^0, x_1^0, x_2^0)$ we perform the following computations (where the successive steps are dependent on the successful completion of the previous ones):

1. Run Newton's method with $(x_0^0, x_1^0, x_2^0)$ as initial guess to find an equilibrium solutions to the Hill model with $s = 0$.

2. If Newton's methods converges to an equilibrium solution $(x_0, x_1, x_2)$ check if it satisfies the condition $0 < x_2 \leq \min \theta_{*,2}$ ($\max \theta_{*,2} < x_2$ for descending hysteresis).

3. If the above condition is satisfied we use the solution $(x_0, x_1, x_2)$ and $s = 0$ as the initial point for a pseudo arclength continuation method to compute a curve of equilibria from $s = 0$ up to $s = 4$.

4. During the continuation of the equilibria we identify saddle-node bifurcations by monitoring the sign of the determinant.

5. If during the continuation of the equilibria we get an even number of saddle-node bifurcations and for each bifurcation the value of $x_2$ just before the bifurcation point is smaller (larger for descending hysteresis) than the value of $x_2$ just after the bifurcation point, then we declare this a hysteretic curve.

If we get a hysteretic curve for at least one of the random initial guesses we declare the sampled parameter point used for the computations as a *hysteretic parameter point*.

The *hysteresis score* of a set of sampled parameter points is the percentage of hysteretic parameter points in the given set of parameter points. For the computations in Tables 1 and 2 we used 1, 000 sampled parameter points (1, 000 curves) for each of the scores. Networks 33, 108, and 4346 used in Table 2 are shown in Fig 10.

## 7 Extending DSGRN capabilities

To be processed by the original DSGRN software [15] a regulatory network was required to satisfy the following conditions:

1. Every node must have an in edge.

2. No repressing self-edges.

3. Every node must have an out edge.

These assumptions are too restrictive as they remove a tremendous number of potentially interesting regulatory networks. The current version of DSGRN [6] overcomes these constraints as indicated below.

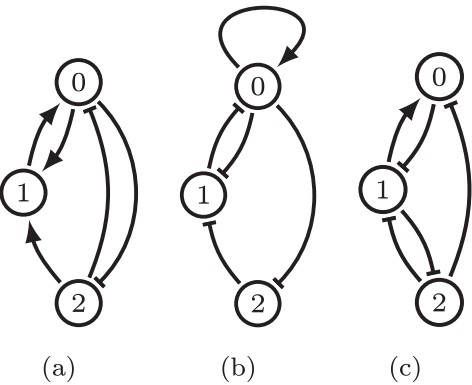

**Fig 10.** (a) Regulatory network 33. (b) Regulatory network 108. (d) Regulatory network 4346.

## 7.1 No in edges

As an example consider network 13 or 14 of Fig 9 where node 1 has an out edge, but no in edges. In general if node $n$ has no in-edges the $n$-th component of (5) reduces to

$$-\gamma_n x_n + \beta$$

and derivation of the parameter regions proceeds as usual (though the computation is trivial) [17].

## 7.2 Node with a self repressing edge

We begin by quickly surveying how DSGRN produces a state transition graph (STG) that is a representative for the dynamics. For more details see [15]. Consider a regulatory network with $N$ vertices. Let $E(n)$ denote the number of out edges from node $n$. Thus, the set of threshold values associated with the $n$-th node is $\Theta(n) \coloneqq \{\theta_{m_k,n}|k = 0,\dots,E(n)-1$ where we assume that $0 < \theta_{m_k,n} < \theta_{m_{k+1},n}$. The complement of the set of hyperplanes $x_n = \theta_{m_k,n}$, $n = 0,\dots,N-1$ defines a collection of open cubical subsets of $(0,\infty)^N$. We refer to these sets as *top cells*. We index these cells by $\mathcal{K} \coloneqq \prod_{n=0}^{N-1}\{0,1,\dots,E(n)\}$ where $(j_0,\dots,j_{N-1})$ is the cell containing points $x$ satisfying $\theta_{m_j,n} < x_n < \theta_{m_{j+1},n}$ with the convention that $\theta_{m_0,n} = 0$ and $\theta_{m_{E(n)+1},n} = \infty$. The boundaries of the top cells are called *walls* and each *interior* wall, i.e. a wall contained in $(0,\infty)^N$, is a subset of a hyperplane $x_n = \theta_{m_k,n}$. Furthermore, each such wall is the boundary element of exactly two top cells and we use this fact to index the walls by $\mathcal{W} \coloneqq \{(\kappa_0,\kappa_1)\} \subset \mathcal{K} \times \mathcal{K}$ where the pair $(\kappa_0,\kappa_1)$ indicates the wall whose two top cells are indexed by $\kappa_0 = (i_0,\dots,i_{N-1})$ and $\kappa_1 = (j_0,\dots,j_{N-1})$. We refer to $i_0,\dots,i_{N-1}$ as the coordinates of $\kappa_0$. Observe that this allows us to adopt the following convention: the wall indexed by $(\kappa_0,\kappa_1)$ is a subset of the hyperplane $x_n = \theta_{m_k,n}$ if and only if $i_n = k-1$, $j_n = k$, and $i_\ell = j_\ell$ for $\ell \neq n$. If we wish to emphasize this information we write $(\kappa_0,\kappa_1)_{k,n}$ Consider $\kappa_0$ and $(\kappa_0,\kappa_1)_{k,n}$. Referring to (5) we define $(\kappa_0,\kappa_1)$ to be *repelling* or *absorbing* (in [15] they are referred to as incoming and outgoing, respectively) with respect to $\kappa_0$ if

$$-\gamma_n \theta_{m_k,n} + \Lambda_n(x) < 0 \quad \text{or} \quad -\gamma_n \theta_{m_k,n} + \Lambda_n(x) > 0, \tag{7}$$

respectively, for $x$ in the top cell indexed by $\kappa_0$. The opposite set of inequalities are used to define repelling and absorbing with respect to $\kappa_1$.

Observe that the indexing of top cells $\mathcal{K}$ only depends on the ordering of the thresholds, not their numerical values. Thus, $\mathcal{K}$ is a purely combinatorial object. Nevertheless, based on the motivating geometry we say that $\kappa$ and $\kappa'$ are *adjacent* if all their coordinate values are the same except for one coordinate and in that coordinate they differ by exactly one. It is only in (7) that the value of the thresholds plays a role. The decomposition of parameter space is chosen such that for each region of the decomposition the inequalities of (7) are preserved.

As is described in [15] if there are no repressive self-edges in the regulatory network, then given $(\kappa_0, \kappa_1)$ the options are:

A1. $(\kappa_0, \kappa_1)$ is absorbing with respect to $\kappa_0$ and repelling with respect to $\kappa_1$;

A2. $(\kappa_0, \kappa_1)$ is repelling with respect to $\kappa_0$ and absorbing with respect to $\kappa_1$;

A3. $(\kappa_0, \kappa_1)$ is repelling with respect to both $\kappa_0$ and $\kappa_1$.

According to these three options the classical DSGRN defines an edge $\kappa_0 \to \kappa_1$ in case **A1**, an edge $\kappa_1 \to \kappa_0$ in case **A2**, and no edge between $\kappa_0$ and $\kappa_1$ in case **A3** (see Fig 11). We can view this as suggesting that the absorbing direction dictates how one top cell is mapped to a neighboring top cell. Performing this computation over all of $\mathcal{W}$ produces the STG. Observe that $\mathcal{K}$ is the set of vertices of the STG and that edges only exist between adjacent elements of $\mathcal{K}$.

If there is a repressive self-edge in the regulatory network, then it is possible that $(\kappa_0, \kappa_1)$ is absorbing with respect to both $\kappa_0$ and $\kappa_1$. The naive response is to introduce edges $\kappa_0 \to \kappa_1$ and $\kappa_1 \to \kappa_0$, but this suggests recurrent dynamics where it may not exist. Thus, this case was not considered in the classical DSGRN.

However, based on [15] (see in particular Section 4.2) we claim the results:

R1. Consider $(\kappa_0, \kappa_1)$ indexing a wall contained in $x_n = \theta_{n,n}$. Then, $\Lambda_i(x) = \Lambda_i(\bar{x})$ for all $i \neq n$ and for any $x \in \kappa_0$ and $\bar{x} \in \kappa_1$.

R2. In addition, consider $(\kappa'_0, \kappa'_1)$ indexing a wall contained in $x_n = \theta_{n,n}$, such that $(\kappa_0, \kappa'_0)$ and $(\kappa_1, \kappa'_1)$ are indices for walls, i.e. $\kappa_0$ and $\kappa'_0$ are adjacent as are $\kappa_1$ and $\kappa'_1$. If $(\kappa_0, \kappa'_0)$ is repelling (absorbing) with respect to $\kappa_0$ or $\kappa'_0$, then $(\kappa_1, \kappa'_1)$ is repelling (absorbing) with respect to $\kappa_1$ or $\kappa'_1$.

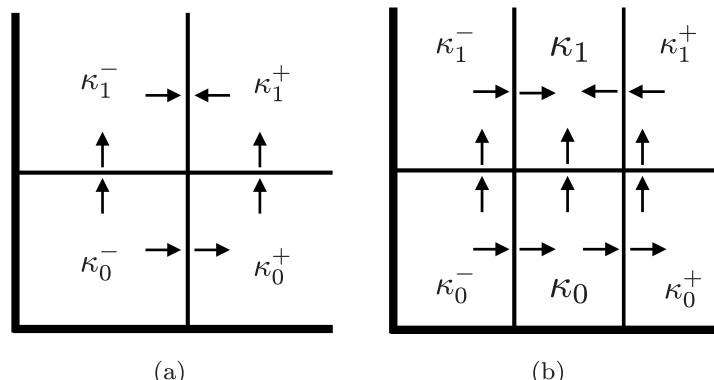

(a)                           (b)

**Fig 11.** (a) A potential original DSGRN complex $\mathcal{K}$ where the vertical threshhold is associated with a self-repressing edge. An arrow going from a top cell $\kappa_i$ to a wall indicates an absorbing wall and an arrow from a wall to a top cell indicate a repelling wall. (b) Portion of the refined DSGRN complex $\mathcal{K}^\dashv$. The cells $\kappa_0$ and $\kappa_1$ are the additional cells in the refined complex. The wall labelling in (b) induces the edges $\kappa_0^- \to \kappa_0$, $\kappa_0^- \to \kappa_1^-$, $\kappa_0 \to \kappa_0^+$, $\kappa_0 \to \kappa_1$, $\kappa_1^- \to \kappa_1$, and $\kappa_1^+ \to \kappa_1$ in the STG on $\mathcal{K}^\dashv$.

We resolve the issue of a self-edge by expanding $\mathcal{K}$. Fix a region of parameter space. This implies that the inequalities (7) are fixed. Define

$$\mathcal{K}^{\dashv} := \prod_{n=0}^{N-1}\{0,\ldots,k,\ldots,E^*(n)\}$$

where $E*(n) = E(n) + 1$ if there is a self-repressing edge to node $n$ and $E*(n) = E(n)$ otherwise. If there is a self-repressing to node $n$ then the threshold corresponding to this edge is $\theta_{m_k,n}$ and we indicate this by denoting $k_-^* = k$ and $k_+^* := k + 1$. In this case it is possible to have a wall $(\kappa_0, \kappa_1)_{k,n}$ such that $(\kappa_0, \kappa_1)$ is absorbing with respect to both $\kappa_0$ and $\kappa_1$. Again based on [15] this is only possible for the $k$ corresponding to the $k_\pm^*$ above. To define the STG we need to consider adjacent cells in $\mathcal{K}^{\dashv}$, i.e., the set $\mathcal{W}^{\dashv} := \{(\kappa_0, \kappa_1)\} \subset \mathcal{K}^{\dashv} \times \mathcal{K}^{\dashv}$ where again it is assumed that a single coordinate of $\kappa_1$ is larger than the coordinate in $\kappa_0$. Let $(\kappa_0, \kappa_1) \in \mathcal{W}^{\dashv}$. If neither $\kappa_0$ nor $\kappa_1$ contain a $k_\pm^*$ as a coordinate, then we use the classical DSGRN rules based on **A1**–**A3**. Thus, we only need to consider $(\kappa_0, \kappa_1) \in \mathcal{W}^{\dashv}$ where either $\kappa_0$ or $\kappa_1$ contains $k_\pm^*$ as a coordinate. Consider $(\kappa_0, \kappa_1)_{k_-^*,n}$. Then the classical DSGRN rules apply to determine whether $(\kappa_0, \kappa_1)$ is absorbing or repelling with respect to $\kappa_0$. If $(\kappa_0, \kappa_1)$ is absorbing (repelling) with respect to $\kappa_0$, define $(\kappa_0, \kappa_1)$ is repelling (absorbing) with respect to $\kappa_1$. Consider $(\kappa_0, \kappa_1)_{k_+^*,n}$. Then the classical DSGRN rules apply to determine whether $(\kappa_0, \kappa_1)$ is absorbing or repelling with respect to $\kappa_1$. If $(\kappa_0, \kappa_1)$ is absorbing (repelling) with respect to $\kappa_1$, define $(\kappa_0, \kappa_1)$ is repelling (absorbing) with respect to $\kappa_0$. Now assume that a $k_\pm^*$ is a coordinate of both $\kappa_0$ and $\kappa_1$ and hence we need to consider $(\kappa_0, \kappa_1)_{\ell,n'}$. Once again there are three cases to consider $\ell = k_-'^*$, $\ell = k_+'^*$, or the $\ell$-th threshold is not associated with a repressive self-edge. Classical DSGRN does not apply for determing absorbing and repelling in any of these cases. To determine this consider $\kappa_0^\pm, \kappa_1^\pm \in \mathcal{K}^{\dashv}$ such that (see Fig 11)

$$(\kappa_0^-, \kappa_1^-), (\kappa_0^+, \kappa_1^+), (\kappa_0^-, \kappa_0), (\kappa_0, \kappa_0^+), (\kappa_1^-, \kappa_1), (\kappa_1, \kappa_1^+) \in \mathcal{W}^{\dashv}.$$

Note that while $\kappa_0^\pm$ or $\kappa_1^\pm$ must exist, it is possible that only one pair exists. Also observe that classical DSGRN applies to the pairs $\kappa_0^\pm$ and $\kappa_1^\pm$ and thus absorbing and repelling of $(\kappa_0^-, \kappa_1^-)$ and $(\kappa_0^+, \kappa_1^+)$ is determined. If both $\kappa_0^\pm$ and $\kappa_1^\pm$ exist, then by **R2** $(\kappa_0^-, \kappa_1^-)$ is absorbing/repelling with respect to $\kappa_0^-$ ($\kappa_1^-$) if and only if $(\kappa_0^+, \kappa_1^+)$ is absorbing/repelling with respect to $\kappa_0^+$ ($\kappa_1^+$). We define $(\kappa_0, \kappa_1)$ to be absorbing/repelling with respect to $\kappa_0$ ($\kappa_1$) in accordance with $(\kappa_0^-, \kappa_1^-)$ or $(\kappa_0^+, \kappa_1^+)$.

## 7.3 Node without an out-edge

DSGRN uses the thresholds corresponding to the out-edges of each node to construct the cubical complex $\mathcal{X}$ decomposing the phase space. For this reason the original DSGRN does not allow for nodes in the network without at least one out-edge [15]. We address this limitation in the following way. We treat a node without out-edges as if it had one single out-edge. In particular, we use the parameter factor graph of a node with a single out edge in the construction of the parameter graph for this node. Hence if $x_i$ is a node without an out-edge, in the parameter decomposition for this node there is a threshold $\theta_{\emptyset,i}$ that is not associated to any edge in the network. Using this approach we have at least one threshold for every node and can construct the cubical complex $\mathcal{X}$ as it is done in the original DSGRN. The threshold $\theta_{\emptyset,i}$ is only used to determine the cubical complex $\mathcal{X}$ at the node $x_i$ and it does not affect the other nodes of the network. In the DSGRN output of parameter inequalities this threshold is displayed as $\top[x_i \rightarrow]$.

## Acknowledgments

The authors thank Bree Cummins for helpful discussions.

## Author Contributions

**Conceptualization:** Marcio Gameiro, Tomáš Gedeon, Shane Kepley, Konstantin Mischaikow.

**Formal analysis:** Marcio Gameiro, Tomáš Gedeon, Shane Kepley, Konstantin Mischaikow.

**Funding acquisition:** Tomáš Gedeon, Konstantin Mischaikow.

**Investigation:** Marcio Gameiro, Tomáš Gedeon, Shane Kepley, Konstantin Mischaikow.

**Methodology:** Marcio Gameiro, Tomáš Gedeon, Shane Kepley, Konstantin Mischaikow.

**Software:** Marcio Gameiro, Shane Kepley.

**Validation:** Marcio Gameiro, Tomáš Gedeon, Shane Kepley, Konstantin Mischaikow.

**Writing – original draft:** Marcio Gameiro, Tomáš Gedeon, Shane Kepley, Konstantin Mischaikow.

**Writing – review & editing:** Marcio Gameiro, Tomáš Gedeon, Shane Kepley, Konstantin Mischaikow.

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
