## [Decision Letter · Decision Letter 0]

22 Dec 2020

Dear Dr. Gameiro,

Thank you very much for submitting your manuscript "Rational design of complex phenotype via network models" for consideration at PLOS Computational Biology.

As with all papers reviewed by the journal, your manuscript was reviewed by members of the editorial board, but in this case only one independent reviewer has submitted a written review. We are taking the unusual step of proceeding with an initial decision on the basis of a single review in order to expedite the process during a time when reviewing resources are stretched and because the review itself is from a known expert in this field who has chosen to reveal his identity. The reviewer has done an excellent job of highlighting the significance and potential interest of this submission, and I concur with his evaluation. He has, however, also brought up a number of important issues that should be addressed in a revised manuscript. Please pay particular attention to his suggestions for improving the presentation of the methodology, which in both his opinion and mine is not currently in a form that it can be understood by a general audience. I found his suggestions for a tutorial style presentation to be very appealing. In light of the reviews and these considerations, we would like to invite the resubmission of a significantly-revised version that takes them into account.

We cannot make any decision about publication until we have seen the revised manuscript and your response to the reviewers' comments. Your revised manuscript is also likely to be sent to additional reviewers for further evaluation.

Sincerely,

James R. Faeder

Associate Editor

PLOS Computational Biology

Mona Singh

Methods Editor

PLOS Computational Biology

Reviewer's Responses to Questions

**Comments to the Authors:**

Reviewer #1: Attached

**Have all data underlying the figures and results presented in the manuscript been provided?**

Reviewer #1: Yes

PLOS authors have the option to publish the peer review history of their article (what does this mean?). If published, this will include your full peer review and any attached files.

Reviewer #1: **Yes: **Leon Glass
---

## [Decision Letter · Decision Letter 1]

29 Apr 2021

Dear Dr. Gameiro,

Thank you very much for submitting your manuscript "Rational design of complex phenotype via network models" for consideration at PLOS Computational Biology. As with all papers reviewed by the journal, your manuscript was reviewed by members of the editorial board and by an independent reviewer. The reviewer appreciated the attention to an important topic and your replies to his previous review. Based on the review, we plan to accept this manuscript for publication, but would ask that you consider modifying the manuscript according to the reviewer's recommendations. Once we receive your reply and revised manuscript we will proceed quickly to process your manuscript for publication.

Sincerely,

James R. Faeder

Associate Editor

PLOS Computational Biology

Mona Singh

Methods Editor

PLOS Computational Biology

[LINK]

Reviewer's Responses to Questions

**Comments to the Authors:**

Reviewer #1: In this revised version, the authors have made several changes, primarily in Sections

4.1-4.4. The changes make the manuscript somewhat more self-contained though it will

still be rough going for the person who does not know this technique but would like to

understand how it can be used. I still have some issues with the revised manuscript.

1. Symmetry. Networks 1-4 all have the same state transition diagrams (an

example of which is now given in Fig. 6) and all have the same hysteresis scores

in Fig. 8. From a computational perspective, and a dynamics perspective these

seem to be the same network as classified by the symmetry under the 3-cube as

demonstrated initially in Glass (1975), Edwards and Glass (2000). The added

sentence in response to my earlier comment should be expanded to clarify so

that a clever reader who recognizes that the symmetry may be relevant would

realize that some aspects of symmetry in these networks have already been

investigated.

2. Multiple inputs to a node. The state transition diagrams will be very different

depending on how multiple inputs combine (e.g. for two positive inputs as an

AND or an OR function). Are the hysteresis scores independent of the type of

interaction? This would seem highly unlikely to me, but a clear statement about

the dependence of the hysteresis score on the implied function for multiple inputs

would be useful.

3. Dynamics of natural systems with hysteresis. As I pointed out in my first review,

there are several earlier papers identifying natural and man-made networks with

multiple steady states and hysteresis. It is great that the authors are collaborating

with synthetic biologists to generate robust networks based on their insights, but I

still believe that the authors should at least mention some of the earlier papers

that have demonstrated or analyzed multiple steady states and indicate how they

fit into the current scheme. For example, the current work would be strengthened

if the circuits for plant development from Elena Alvarez-Buylla’s group conformed

to their robust networks. If the networks did not conform, the authors could

predict that additional control elements should be present (a prediction almost

certain to be correct!).

4. Robustness. The authors now mention two earlier papers that deal with

robustness, but do not give a clear indication of how the current formulation

relates to the earlier ones. Although the current paper deals with hysteresis,

presumably it could also deal with limit cycle dynamics. If so, would the earlier

concept of stable attracting paths in the state transition diagram leading to robust

dynamics match the formulation of the current work?

The authors should correct the spelling of Kauffman’s name. Also, missing Section

numbers need to be corrected.

It would not be too difficult to implement any changes in response to these comments

and I would leave it up to the authors whether they wish to do so, or not.

Leon Glass

**Have the authors made all data and (if applicable) computational code underlying the findings in their manuscript fully available?**

Reviewer #1: Yes

PLOS authors have the option to publish the peer review history of their article (what does this mean?). If published, this will include your full peer review and any attached files.

Reviewer #1: **Yes: **Leon Glass

Figure Files:

Data Requirements:

Reproducibility:

References:

---

## [Editor Report · Decision Letter 2]

17 Jun 2021

Dear Dr. Gameiro,

We are pleased to inform you that your manuscript 'Rational design of complex phenotype via network models' has been provisionally accepted for publication in PLOS Computational Biology.

Best regards,

James R. Faeder

Associate Editor

PLOS Computational Biology

Mona Singh

Methods Editor

PLOS Computational Biology

---

## [Editor Report · Acceptance letter]

23 Jul 2021

PCOMPBIOL-D-20-02006R2 

Rational design of complex phenotype via network models

Dear Dr Gameiro,

I am pleased to inform you that your manuscript has been formally accepted for publication in PLOS Computational Biology. Your manuscript is now with our production department and you will be notified of the publication date in due course.

With kind regards,

Agota Szep
